# Development and Mechano-Chemical Characterization of Polymer Composite Sheets Filled with Silica Microparticles with Potential in Printing Industry

**DOI:** 10.3390/polym14163351

**Published:** 2022-08-17

**Authors:** Sidra Siraj, Ali H. Al-Marzouqi, Muhammad Z. Iqbal

**Affiliations:** Chemical Engineering Department, College of Engineering, United Arab Emirates University, Al Ain 15551, United Arab Emirates

**Keywords:** sustainability, silica, sand, polymer, composite, synthetic, sheets

## Abstract

Polymer composite sheets using a low-cost filler (local natural sand) and polymer (high-density polyethylene, HDPE) as a replacement of the traditionally used wood-fiber-based sheets for paper-based applications were developed. The sand/polymer composite sheets were prepared by melt extrusion in a melt blender followed by compression molding. The effects of varying particle size, concentration, and the use of a compatibilizer (polyethylene-grafted maleic anhydride) was studied on the mechano-chemical performance properties of the composite sheets such as morphology, thermal and mechanical properties, and wettability characteristics used in the printing industry. In terms of thermal stability, filler (sand) or compatibilizer addition did not alter the crystallization, melting, or degradation temperatures significantly, thereby promoting good thermal stability of the prepared sheets. Compatibilization improved anti-wetting property with water. Additionally, for the compatibilized sheets prepared from 25 µm sand particles, at 35 wt%, the contact angle with printing ink decreased from 44° to 38.30°, suggesting improved ink-wetting performance. A decrease in the elastic modulus was also observed with the addition of the compatibilizer, with comparable results to commercial stone paper. Results from this study will be considered as a first step towards understanding compatibility of local natural sand and polymers for paper-based application.

## 1. Introduction

Traditionally, paper is derived from wood by pressing together moist fibers of cellulose pulp followed by drying. Global use of papers for various applications, such as printing papers, bags, posters, cutlery, etc., is also adding a toll on the environment as it requires a huge amount of water to wash away the initial pulp as well as requires chemicals to bleach such products, thereby also increasing wastewater and sludge disposal problems.

Additionally, increasing population has increased the global production and use of paper and paper products exponentially. An average global growth rate of paper production was almost 3.6% in the 1980s and approximately 3.3% in the 1990s, currently reaching up to a massive 400 million tons per year requirement [1]. On the other hand, increasing use of plastics has become a major cause of concern as it leads to drastic environmental issues due to its non-eco-friendly disposal [2]. An estimate of about 2500 million metric tons (MMT) of plastics are currently in use. It is also reported that over the past 65 years, the cumulative amount of waste generated from primary and recycled plastic waste adds up to a staggering amount of ~6000 MMT. Of this waste, approximately 12% is incinerated and 9% is recycled. Furthermore, over 60% of all plastics produced are directly discarded in nature (marine or in landfills) and continue to accumulate to this date [3]. Figure 1 illustrates a schematic of the worldwide production and disposal of plastics in MMT for the last 65 years. At the current consumption rate of plastics, the Earth will hold an estimate of 3.3 billion tons of plastic by 2050 [4].

Accordingly, both issues require more innovative methods to help tackle current environmental problems as well as benefit the society in a more sustainable way. Hence, the development of synthetic “paper” came into existence. Synthetic paper or sheet produced from thermoplastic polymers (virgin or recycled) and fillers and additives form a thin composite film that can be used in a variety of paper-based applications. Since the synthetic composite sheet is a “tree-free” product, it will contribute towards reducing the environmental impact of the paper industry, and its potential can be even increased when using recycled plastics [5]. 

The detrimental effects of manufacturing paper and paper-based products from wood on the environment have led to the commercialization of synthetic papers on a global scale. Taiwan Lung Meng industries was the first to commercialize a synthetic paper named “stone paper”, which is manufactured using minerals dispersed in polyolefins [6]. The stone paper, which is used to produce a variety of paper-based products such as bags, calendars, labels, etc., has been given several other commercial names, such as Rockstock [7] and mineral paper, and is eco-friendly with the main composition being a varying ratio of a polymer and suitable filler. 

Synthetic “paper” or filler-polymer composite sheet has properties similar to the traditional cellulose-based paper [8]. It is developed by extruding polyolefin films which may be multilayered and are filled with inorganic particles, more commonly known as fillers. A large number of inexpensive inorganic particulate materials have been widely used as fillers to enhance the thermal and mechanical properties of polymers such as calcium carbonate (CaCO_3_), glass, titanium dioxide (TiO_2_), and silica (SiO_2_) in paper-based applications [9,10,11]. Out of the mentioned fillers, CaCO_3_ has been extensively studied and used for paper-based applications as it can provide the properties required such as opaqueness and printability, as well as increase the mechanical properties [2,12]. However, an increase in such a filler leads to an increase in its brittle nature [10,11].

The Middle East, including the UAE, similar to any other onshore country, has naturally abundant sand (silica). The sand available in the UAE varies with geological position in composition, color, size, and surface texture. Due to its high thermal stability, high durability, and unreactive nature, silica is used in various fields such as in 3D printing, construction, textile, electronic applications, etc. [13,14,15,16,17,18,19,20]. Using UAE’s sand in writing paper applications can be cost-effective and produce more eco-friendly sheets compared to wood-based paper. 

In general, the existing literature indicates the extensive development and commercialization in the production of synthetic composite sheets and its potential replacement for wood-based sheets. The literature represents increased mechanical and durable properties as well as increased thermal stability and degradation, signifying its good potential for such a product [13,14,20,21,22,23]. The literature also shows the use of sand in polymers to improve properties in applications such as construction, waste management, and recycling applications [24,25,26,27,28]; however, not much study has been conducted on using natural sand (silica) as a filler to develop sheets for printing applications.

Therefore, the main objective of this research is to utilize the UAE’s sand in manufacturing synthetic composite sheets for writing/printing applications using melt extrusion technique followed by compression. The physiochemical properties of developed composite sheets are assessed and compared with commercial stone paper and regular A4 paper for writing applications. The goal is to develop paper-like composite sheets using a natural resource to contribute towards a more sustainable approach of paper production. The presented solution contributes to using sand as a raw material in a useful way and thereby widens its potential for industrial use [29]. Additionally, the novelty of this work lies in attempting to utilize the raw local sand as a potential filler in a composite material that can have a commercial-scale application, such as in the printing industry, which can potentially grow as an alternative to the traditional wood-based fiber sheets which are extensively used.in numerous applications.

## 2. Materials and Methods

Sand sample was collected from Ras Al-Khaimah, UAE (25.6741° N, 55.9804° E). The local sand was classified into 3 types (carbonates, silicates, and free silica, which is in the form of detrital quartz). The actual relative abundance in terms of the mineral content was analyzed to be about 47 wt% silicates, 26 wt% carbonates, and 14 wt% quartz. The mineral composition of SiO_2_ was analyzed to be 36.93 wt% [30]. All chemicals used, unless stated otherwise, were supplied by Sigma Aldrich, Germany and were of appropriate purity for application in this study. High-density polyethylene (HDPE) pellets (ρ = 0.98 g/cm^3^) were purchased from Sigma-Aldrich. PE-g-MA (ρ = 1.8 g/cm^3^) pellets were also purchased from Sigma-Aldrich. Sand from the UAE was collected and ground to finer particles using heavy-duty grinders. A sieve (200 mm diameter, 25 microns aperture, stainless steel mesh) was further used to refine the fine powder to a particle size of 25 µm. Grinding sand to 5 µm was performed by Retsch, Haan, Germany.

### 2.1. Experimental Procedure

The sand/polymer composite sheets were prepared via melt-blending, followed by compression molding. For all the experiments, American Standard Test procedures were adopted.

### 2.2. Synthesis of Sand/Polymer Composite Material via Melt-Blending

Weighed amounts of sand at two different particle sizes (5 and 25 µm) and different fractions (0 wt%, 20 wt%, 35 wt%, and 50 wt%) were mixed with HDPE and melt-blended in a twin-screw extruder (MiniLab HAAKE Rheomex CTW5, Germany) for 15 min at 170 °C and at a screw speed of 100 rpm. These conditions were selected after several trials of varying the time, temperature, and rpm for the process. The total feed amount was kept constant at 4 g as required by the extruder setup [23]. The control sample was prepared by blending pure HDPE using the same conditions as in numerous studies [31,32,33,34]. For the samples prepared with the addition of the compatibilizer (C), the composition of the compatibilizer was kept constant at a filler-to-compatibilizer ratio of 2:1. Compatibilizers are used as additives to enhance the properties of the fabricated polymer composites prepared by various techniques. Studies report polyethylene grafted with maleic anhydride (PP-g-MA) as a compatibilizer to increase the dispersion of the clay in the nonpolar polymer matrices and improved properties [14], which is why it was chosen. PE-g-MA and its ratio was also chosen after several trials of varying the compatibilizer type and amount. Table 1 represents chosen weight percentages for the three components for the preparation of the sand/polymer composite sheets.

#### Synthesis of Sand/Polymer Composite Sheets via Compression Molding

The compounded composites were chopped into small pieces (~1 g) and compressed (hot-pressed) for 10 min using a Carver’s press (Carver™ Lab Presses) under 5000 psi pressure at 170 °C to prepare the flat sand/polymer composite sheets. Sand/polymer composite sheets were prepared from two different sizes (25 µm and 5 µm) of sand as the filler. The same sets were prepared with the addition of the compatibilizer as well. Figure 2 and Figure 3 show images for sand/polymer composite sheets prepared from 25 µm sand particles without and with compatibilizer, respectively, whereas Figure 4 and Figure 5 represent the same for sheets prepared with 5 µm, respectively. All the sheets were roughly 1 mm thick and 100 mm in diameter. It could be visually seen that increasing filler addition made the sheets darker in appearance. However, addition of compatibilizer produced sheets that were relatively lighter in appearance and softer. Furthermore, based on a visual inspection, it can be said that the dispersion was random in all sheets, with a more homogeneous dispersion in 20 wt% and 35 wt%, and a non-homogeneous dispersion was observed in both cases of 50 wt% filler before and after compatibilization. The overall process is illustrated in Figure 6. A similar process was followed with the addition of compatibilizer.

### 2.3. Characterization

#### 2.3.1. Scanning Electron Microscope (SEM)

A JEOL/EO scanning electron microscope (SEM), operated at 2 kV, spot size of 40, was used to image the sand of both 25 µm and 5 µm, and neat HDPE at their surface. To improve conductivity and quality of image, samples were coated with Au/C using a vacuum sputter coater. Likewise, SEM analysis was used to observe the surface morphology of the selected composite sheets. Selected composite sheets were placed on an aluminum pin-mount adapter using double-sided carbon tape and then were sputter-coated with gold using a sputter-coater to avoid electrostatic charging during examination. The SEM was operated at high vacuum mode with an acceleration voltage of 15 kV and the images were acquired.

#### 2.3.2. X-ray Diffraction (XRD)

X-ray diffraction (XRD) was performed on sand samples (within 2θ = 5–40°) using a Panalytical X-ray diffraction system (X’Pert3 Powder, Malvern Panalytical, Denver, CO, USA) to confirm the presence of SiO2, which is the major component of sand, as reported in the literature [35]. XRD was also performed on HDPE after the addition of the compatibilizer to determine any alteration in the characteristic peaks of HDPE. Similarly, XRD was performed for selected composite sheets to observe any shifts or alterations due to filler and compatibilizer addition.

#### 2.3.3. Differential Calorimetry Analysis (DSC)

The melting temperatures (Tm) and the crystalline temperatures (Tc) of the control sample, i.e., the neat HDPE sheet and the prepared synthetic composite sheets, as well as the compatibilized composite sheets, were determined using modulated differential scanning calorimetry (Discovery DSC 25, TA Instruments, New Castle, DE, USA). About 5–10 mg of sample was heated from 20−180 °C at a heating rate of 40 °C/min to remove thermal history of the polymer. Once the thermal history was eliminated, all samples were cooled from 180 °C to room temperature (20 °C) at 10 °C/min for recording the crystallization temperature (Tc) followed by subsequent heating scans from 20–180 °C at 10 °C/min to record the melting temperatures (Tm). All experiments were carried out under inert nitrogen atmosphere. The same was also repeated for stone paper and regular A4 paper for comparative analysis.

#### 2.3.4. Thermogravimetric Analysis (TGA)

Thermogravimetric analysis of neat HDPE, the prepared composite sheets, and stone paper, as well as for regular A4 paper, was carried out using TGA (Q500 series, TA Instrument). For each experiment, a sample weight of 6.0 mg (±1.0) was used for thermogravimetric analysis. The heating rate was controlled at 10, 15, 20, and 25 °C/min from 25 °C to 900 °C, using nitrogen as a carrier gas at 20 mL/min. During the thermo-decomposition process, the initial weight was recorded continuously as a function of temperature and time. The derivative (DTG) curve was also plotted for the weight loss of sample per unit time.

#### 2.3.5. Tensile Test

Durability of the prepared composite sheets was tested using mechanical testing. Tensile properties of the composite sheets were determined by the Universal Testing Machine (UTM) using the American Society for Testing and Materials (ASTM)-D 638. Dumbbell-shaped test specimens were prepared from the composite sheets of 35 (l) × 0.5 (w) × 1 (t) mm, and their tensile properties were studied on a Zwick 50 kN. The rate of crosshead motion was set to be 100 mm/min, which was taken from the ASTM D 790 standard. The same was also repeated for stone paper and regular A4 paper for comparative analysis.

#### 2.3.6. Wettability

Contact angle measurement (θ) was used to determine the wettability of composite sheets which can be determined by the standard ASTM D 2578–17 test. This test assumes that the surface energy of a film (γ) is equal to the surface tension of a liquid that wets the film, without merging, when a drop is placed on the surface via contact angle measurements. Each composite sheet was placed in the Teclis tracker tensiometer equipment setup, and a droplet of liquid (water) was ejected out of the micrometer syringe onto the sample using a gauge needle. Images of the droplet on the surface of the sample were taken, and, using the Teclis tracker software, the angle between a tangent drawn on the liquid droplet and the surface of the composite sheet was calculated as reported in the literature [13].

Figure 7 shows samples for the tangent to the surface drawn on a sheet surface for non-polar, extremely non-polar, and polar surfaces, respectively. The same test was also performed using a non-polar solvent, benzene, a mixture of water-non-polar solvent, as well as pure printing ink (Epson ink 664, black, Carrefour, Al Ain, UAE) to test for the wettability characteristics of the prepared sheets for ink-based applications. The wettability tests were also performed on stone paper and regular A4 paper for comparative study.

#### 2.3.7. Printing Test

Selected composite sheets had no surface treatment and were cleaned with a dry tissue paper. The sheets were then glued on a regular A4 paper, dried for approximately 15 min, and then sent to the printer (RICOH class driver laser printer) as conducted by other studies [2,36]. As a comparison, a printing test was also performed on stone paper and regular A4 paper as well.

#### 2.3.8. Adherence Test

To test the adherence of ink on the prepared sand/polymer sheets, marker pens were used to write on the sheets. After writing on the sheets, the sheets were left to dry for 15 min. A piece of clear adhesive tape (1.5 cm wide) was then applied onto the surface where text was written. Any small bubbles that appeared while applying the clear tape were removed manually, and by using the rolling weight of the hand, the tape was firmly pressed. Consequently, the tape was pulled off each sample, implying that the smaller the amount of ink that was removed by the tape, the better the assessment. The test was repeated twice; once with permanent marker pen and the second time with removable marker pen. The adherence of ink to the selected sand/polymer composite sheet surfaces was assessed qualitatively as excellent, good, regular, or poor, as reported in the literature [13]. The removed tape was placed alongside the sheet to observe the difference. As a comparison, an adherence test was also performed on stone paper and regular A4 paper.

## 3. Results and Discussion

### 3.1. Morphology

In order to determine the size of the ground sand samples, SEM analysis was conducted to obtain an idea of the particle size as well as the shape of the particles. Figure 8a,b show the two sand samples, respectively. It can be clearly seen that the 25 µm sand particles have larger, more irregular-shaped distant particles, whereas the 5 µm sand particles have smaller, finer, and closer particles comparatively. The neat HDPE sheet, with 0 wt% filler, which was processed in the same way as the composite sheets was also analyzed using SEM. Figure 8c shows only a fine structure which is expected in a neat polymer film and is in agreement with the literature [37].

Selected composite sheet samples were also chosen to conduct SEM analysis. At 35 wt% prepared from 5 µm, particles were visibly seen. However, after compatibilization, particles had decreased interparticle distance, suggesting improved dispersion and binding ability [21]. The literature reports good appearance at ~30 wt% for composite sheets as well [2]. Similarly, in the case of 50 wt% and 5 µm particle size, compatibilization shows increased presence of particles, thereby indicating decreased interparticle distance. An increase in the filler accumulation can result in agglomeration of particles, which can lead to a brittle material [13]. SEM images of both sets of composite sheets at 50 wt% are shown in Figure 8d,e, respectively.

### 3.2. XRD Analysis

The successful preparation of the composite sheets with sand as well as with the addition of the compatibilizer (C) was confirmed using XRD analysis. HDPE characteristic peaks at 2θ = ~21.5° and 2θ = ~23.5° were observed, which are supported in the literature as well [38,39]. Addition of the compatibilizer did not alter the characteristic HDPE peaks as shown in Figure 9; however, slight broadening of the peak was observed.

Just as a confirmation, XRD was also performed for selected sand/HDPE composite sheets. Figure 9 also shows that the presence of the 5 µm sand particles and compatibilizer did not alter the characteristic peaks of HDPE, as two distinct diffraction peaks of HDPE were observed for all the prepared sand/polymer composite sheets. As for the sand, two significant characteristic peaks were obtained at 2θ = 21.38° and 23.71°, which are quite comparable to the literature values obtained for precipitated silica as well as sol–gel-produced silicon dioxide, ranging from 21.8° to 23°, respectively [40,41]. The presence of these peaks was noticed for all the prepared samples at values close enough to the 2θ values (with slight shift to the right), showing that the addition of the filler (sand) does not significantly alter the basic structure of the prepared sand/polymer composite sheets at the molecular level as well.

### 3.3. Thermal Analysis

#### 3.3.1. Melting and Crystallization Behavior

DSC analysis was conducted for all sand/polymer composite sheets prepared from both 25 µm sand particles and 5 µm sand particles at the same heating rate (10 °C/min) to understand the crystallization properties of the prepared sand/polymer composite sheets. Figure 10a shows the cooling (first cycle) and heating (second cycle) profiles for the neat HDPE. The observed peak melting temperature (Tm,peak) and peak crystallization temperature (Tc,peak) values for the neat HDPE were ~133.86 °C and ~117.50 °C, respectively, which is also well reported in the literature [38]. A trend of smooth transition temperatures is seen by the neat HDPE, which also shows the absence of any impurity in the sample. 

Figure 10b,c show the cooling and heating profiles for the compatibilized sand/polymer composite sheets prepared from 5 µm sand particles, respectively. The thermal properties for all the sand/polymer composite sheets prepared from 25 µm sand particles and 5 µm sand particles are reported in Table 2.

The obtained DSC results show that the crystallization temperature of HDPE is not influenced due to the addition of filler (sand) or the compatibilizer. All the samples exhibited a single crystallization exotherm and a corresponding melting endotherm. From the DSC thermograms, the onset crystallization temperatures (Tc,onset), peak temperatures for crystallization exotherms (Tc,peak), and melting endotherms (Tm,peak) for neat HDPE, all the prepared sand/polymer composite sheets, as well as stone paper and regular A4 paper, were evaluated. Determining the Tc,onset, Tc,peak and Tm,peak help in defining the processing temperature range of the polymer. Usually, the processing temperature of polymers is ±40 °C from the melting temperature, as reported by various studies [25,35]. Furthermore, by observing the changes that occur in the Tc,peak and Tm,peak, the values of their respective enthalpies can be calculated, which gives a wider idea of how much heat and energy is needed in the manufacturing process of such polymeric sheets.

Slight changes were observed for all the samples in the Tc,onset and Tc,peak in addition to the peak broadening/stretching during crystallization. For the compatibilized composite sheets prepared from 5 µm sand particles, the peak crystallization temperatures (Tc,peak) increased slightly from ~114 °C at 0 wt% to ~115 °C at 50 wt%. As for the peak melting temperatures (Tm,peak), the temperatures decreased slightly from ~133 °C at 0 wt% to ~132 °C at 50 wt%. Moreover, due to similar melting and crystalline temperatures obtained in all the prepared sand/polymer sheets, it can be inferred that filler or compatibilizer addition does not alter the thermal characteristics of HDPE and promotes their good thermal stability, as reported in literature as well [42,43]. The Tc,onset, Tc,peak, and Tm,peak for neat HDPE and for all the prepared sand/polymer composite sheets, as well as stone paper, are presented in Table 2.

The enthalpies of crystallization (ΔHc) and melting (ΔHm) were calculated by integrating the area under the cooling and heating curves, respectively. The percentage of crystallinity (Xc) in the prepared sand/polymer composite sheets was calculated using the following equation:(1)$$Xc\left(\%\right)=\frac{\Delta \mathrm{Hm}}{\Delta H_{100\%}\left(1-\theta \right)}\times 100\%$$
where Xc is the percentage of crystallinity of HDPE, ∆Hm is the melting enthalpy, ∆H_100%_ is the melting enthalpy of a 100% crystalline HDPE, taken as 293 J/g [43], and θ is the mass fraction of the filler (sand). For stone paper, the weight is considered as the manufacturer states, which is 70 wt% [7].

The observed melting enthalpy (ΔHm) in all the prepared sand/polymer composite sheets was always lower than for the pure HDPE (148.77 J/g). Moreover, a general fluctuating value of percent crystallinity (Xc) was observed throughout the prepared sand/polymer composite sheets, ranging from a crystallinity of 40–60%, which is well reported in several filler–polymer composite systems [36,44,45]. This variation is explained by either an insufficient amount of filler (sand) particles at the surface which can cause agglomerates, or an excess accumulation of filler particles which can form a soft layer at the interface present, which tend to decrease the nucleating effect [42].

Moreover, the reduction of the sand/polymer composite melting enthalpies (ΔHm) and crystallinity (Xc) could be further explained by the reduction in the conformational changes available to the macromolecules during crystallization, which is due to the presence of the silica particles in the composites which are not densely packed. According to statistical thermodynamics, particles, in this case, silica, restrict the mobility of macromolecules and reduce the spaces available to be occupied by the macromolecules, thereby restricting the ability to form well-developed crystals. Additionally, the crystalline phase is not densely packed, which results in minimum intermolecular interactions and hence a decrease in the heat of fusion on melting which is also experienced by other HDPE systems such as carbon black/HDPE composites, wood/HDPE composites, and clay/HDPE composites [39,42].

Reduction of ΔHm in industrial terms can be translated to money and power savings during the extrusion/molding process, which encourages a positive attribute for the industrial process. ΔHm, ΔHc, and Xc for all the prepared sand/polymer composite sheets, as well as stone paper, are also presented in Table 2.

#### 3.3.2. Thermal Stability of Composite Sheets

The amount of filler can have a significant impact on the end use properties, for instance, thermal expansion, stiffness, etc., of a final product. The thermal stabilities of the prepared sand/polymer composite sheets were also analyzed by thermal gravimetric analysis (TGA) by observing their onset degradation temperatures (T_d,onset_)_,_ peak degradation temperatures (T_d,peak_), and their onset degradation temperatures at 10% weight loss (T_d,onset @ 10% weight loss_). By determining the T_d,onset_ and T_d,peak_ values, the upper range of processing temperature and the maximum temperature before degradation of the material can be confirmed, respectively. Hence, the range of processing temperature can be optimized to avoid any degradation of the material to occur which proves to be of substantial value for polymeric industrial scale processes. By noticing the changes occurring to the T_d,onset @ 10% weight loss_, the effect of filler percentages and compatibilizer can be evaluated on the initial degradation corresponding to the same weight loss occurring. This can suggest which composition has more impact on the onset of degradation.

The TG curves for 25 µm sand particles and 5 µm sand particles alongside neat HDPE and prepared sand/polymer composite sheets, as well as their compatibilized versions, are shown in Figure 11a–d, respectively. Figure 11 shows that the composites have two degradation steps; the first degradation step is for the polymer, the second degradation is for the sand particles, which is comparable to the neat HDPE, which has only one degradation step, and sand has degradation at higher temperatures [46,47]. HDPE was stable (no significant weight loss) up to 400 °C, confirming that the processing of HDPE and HDPE composites at 170 °C would not degrade the polymer. The complete weight loss was observed near 490 °C, where almost all of the HDPE was burned. The peak degradation temperature (T_d,peak_) for HDPE (obtained from derivative of weight loss curve (DTG)) was observed at ~489 °C. 

The degradation steps became more evident at 50 wt% of filler addition. Additionally, it can be seen that increasing the amount of the filler increases the T_d,onset_ and shifts the thermograms to the right, implying improved thermal stability, which is also reported for another study conducted on a SiO_2_/polymer composite system [47]. Moreover, in the case of sand/polymer composite sheets prepared from 25 µm sand particles, the T_d,peak_ increased very slightly from ~491 °C at 20 wt% to ~492 °C at 35 wt%, and T_d,onset @ 10% weight loss_ increased from 448 °C to 453 °C, respectively. Furthermore, there is also a slight increase in the Tonset for the same set from 20 wt% to 35 wt%, which is also explained in the literature as due to the presence of filler minimizing the permeation of heat [47,48,49].

In addition to that, it can also be observed that the T_d,peak_ after compatibilization were always lower than their corresponding samples prepared without the addition of the compatibilizer (except for sand/polymer sheet prepared at 20 wt% from 5 µm sand particles). This can be explained as due to the presence of acidic groups of maleic anhydride in the compatibilized sheets possibly interacting with some parts of the filler, resulting in slightly faster degradation [50]. For instance, for the compatibilized sand/polymer composite sheets prepared from 5 µm, the T_d,peak_ decreased very slightly from ~499 °C at 20 wt% to ~488 °C at 50 wt%. However, the overall T_d,peak_ remained comparably close enough to the neat HDPE value (489 °C), hence promoting good thermal stability for the prepared sand/polymer composite sheets. A similar trend was observed for the case of sand/polymer composite sheets prepared from 25 µm sand particles.

The differential rate of weight loss (dW/dt) of all the prepared sand/polymer composite sheets was obtained from differential thermogravimetric analysis (DTG) at a set heating rate of 20 °C/min. Figure 11 also shows the DTG plots (as insets) for neat HDPE (0 wt%) and for sand/polymer composite sheets at varying compositions prepared from 25 µm sand particles and 5 µm sand particles without the addition of the compatibilizer and with the addition of the compatibilizer, respectively.

A large fraction of the sand/polymer composite sheets decomposed between 300 °C and 600 °C, and this can be attributed to the decomposition of the HDPE. The thermal decomposition peak between 600 °C and 800 °C was assigned to the decomposition of sand particles which is more visible in samples prepared from 35 wt% composition and 50 wt% composition. 

Both the DSC and TG analyses results indicated that the addition of sand particles of both 25 µm particle size and 5 µm particle size, as well as the addition of the compatibilizer, did not affect the thermal properties of the prepared sand/polymer composite sheets significantly, which promotes good thermal stability for the sheets. Comparable results for thermal stability are reported by numerous studies for various filler–polymer composites [36,51]. Table 3 reports the onset degradation temperatures (T_d,onset_), peak degradation temperatures (T_d,peak_), and onset degradation temperatures at 10% weight loss (T_d,onset @ 10% weight loss_) for all the prepared sand/polymer composite sheets, stone paper, and regular A4 paper for comparison.

### 3.4. Mechanical Properties

The effect of filler and compatibilizer (C) on the elastic modulus and tensile strength for sand/polymer composite sheets prepared from 25 µm sand particles and 5 µm sand particles was studied with the elastic modulus for both sets illustrated in Figure 12a,b, respectively. As seen in Figure 12, at 0 wt% filler the elastic modulus obtained for the pure HDPE sheet was ~1200 MPa with a corresponding yield stress of 35.15 MPa, respectively, which is comparable to the literature [2,45,52]. Generally, a decrease in the elastic modulus was observed with increasing filler concentration from 1298.33 MPa at 20 wt% to 905. 72 MPa at 50 wt% for the sheets prepared from 25 µm sand particles (in Figure 12a). Similar trends were observed in the case of 5 µm, where the elastic modulus dropped from 950.59 MPa at 20 wt% to 887.47 MPa at 35 wt% (except for an increase in the case of 50 wt%), as seen in Figure 12b.

Compatibilization lowered the elastic modulus for all cases compared to their non-compatibilized versions. For instance, compatibilized sand/polymer composite sheets prepared at 35 wt% experienced a sharp reduction in elastic modulus from 1182.23 MPa before compatibilization to 629.95 MPa after compatibilization, and from 887.47 MPa to 687 MPa for sand/polymer composite sheets prepared from 25 µm sand particles and 5 µm sand particles at the same filler composition, respectively (Figure 12a,b). The effect of addition of fillers and compatibilizers on the decreasing value of elastic modulus is observed for many polymer composites and is believed to be caused by the random agglomeration of particles weakening the polymer matrix, causing uneven crystallization leading to their brittle nature, which also contributes to weaker adhesion between filler particles and the polymer matrix [34,47,51,53]. This also contributes to the fact that the compatibilizer (maleic anhydride) is not able to form strong bonds with the composite sheets, leading to a weaker matrix structure, as seen for other blends as well [54,55]. 

Furthermore, with elastic moduli of 603.54 MPa and 687 MPa, the compatibilized sand/polymer composite sheets prepared with 5 µm sand particles at 20 wt% and 35 wt%, respectively, gave results closest to the stone paper, which yielded an elastic modulus of 596.32 MPa. Moreover, another study on composite sheets also suggested that optimum results were obtained from sheets prepared at 30% by weight [2].

The tensile strength did not change significantly with increasing the filler concentration before and after compatibilization, which is reported in the literature as well [56]. However, the tensile strength decreased significantly with the addition of the compatibilizer for both sets of sand/polymer composite sheets prepared from 25 µm sand particles and 5 µm sand particles. Even though compatibilizers are expected to increase the tensile strength, the decreasing trend could possibly be explained due to the compatibilizer limiting the stress transfer and swamping the surface [57]. Furthermore, as increasing the filler decreases the mechanical properties (such as the elastic modulus), the decreased tensile strength can also be a combined effect of the addition of the compatibilizer and filler, as reported in another study on a filler–polymer composite material [53]. 

A wide variation of mechanical properties was observed for the case of sheets prepared at 50 wt%, irrelevant of the particle size, which is also reported to occur in thermoplastic-based films subjected to extensive molecular orientation [2]. For filler content of 50 wt%, the lowest tensile strength values were obtained, thereby suggesting increased brittleness for sheets prepared at higher filler percent by weight, which is also supported by the literature [45]. Due to possible formation of hydrogen bonding, crack propagation at weak phase interfaces can be facilitated, resulting in a lower tensile strength of the blend, which is also reported in the literature [54]. Comparatively, one of the highest tensile strength values (15.66 MPa) was measured for the regular A4 paper, which can be corresponded to its fibrous network structure with strong hydrogen bonding [58]. Table 4 reports all the mechanical properties of sand/polymer composite sheets and compatibilized sand/polymer composite sheets prepared from 25 µm sand particles and 5 µm sand particles, as well as stone paper and regular A4 paper.

### 3.5. Wettability Performance

Surface wettability is one of the most essential properties to determine the use of the material in a specific application, such as in this case for printing-based applications. Wettability is determined based on the contact angle measured on the surface of the material. Typically, any non-polar surface would measure a contact angle of 90° or above. Anything below 90° suggests increased hydrophilicity with the liquid on the surface and results in increased absorption of the liquid in contact. Increasing the filler concentration increased the contact angle slightly from 86.62° at 20 wt% to 94.72° at 50 wt% and 88.88° at 20 wt% to 94.6° at 50 wt% for sand/polymer composite sheets prepared from 25 µm sand particles and 5 µm sand particles, respectively, suggesting improved anti-wetting performance with water for both particle sizes. These results indicate that the nonpolar, hydrophobic blocks of the compatibilizer units may have been arranged on the surface of the composite sheets, as reported for other blends in the literature as well [54]. Contact angles of above 90° were obtained for all the compatibilized sheets. Figure 13 illustrates this data.

In the case of the non-polar solvent, benzene, decreased contact angles were obtained (ranging between 20–35°) for all the sand/polymer composite sheets prepared, suggesting their absorption ability to a certain extent if they were to come into contact with any organic solvents. This can be excepted, as the surface is mainly non-polar and any contact with a non-polar liquid would promote the widely known “like dissolving in like” phenomena. Since maleic anhydride is an organic compound, the oxygen in its functional group binds to the chemical structure of benzene. Values of ~90° were obtained when the sheets were tested with a water–benzene mixture. In addition to that, regular A4 paper also exhibited a relatively higher value of contact angle (82.27°) with water–benzene mixture.

Contact angle values of lower than 45° were obtained with printing ink. Irrespective of the particle size used, a general trend of decreased contact angles was observed with the addition of the compatibilizer compared to their non-compatibilized versions, indicating improved ink-wetting performance. For example, as seen in Figure 13 for sand/polymer composite sheets prepared from 25 µm sand particles, for both at 35 wt% and 50 wt%, the contact angles decreased from ~44° to ~38°. Similar contact angles for ink-wetting have been reported [59]. Moreover, the values obtained were relatively close to the commercial stone paper, which resulted in a contact angle of 32.50° with printing ink. Contact angles below 50° correspond to surface free energies of below 45 dyne/cm, which is preferred for printing [13,60]. A contact angle of 0° was obtained for regular A4 paper with printing ink, which is expected due to its increased hydrophilic surface. The wettability performances with printing ink for all the sand/polymer composite sheets prepared from 25 µm sand particles and 5 µm sand particles are illustrated in Figure 13.

Table 5 summarizes all the measured contact angles for the prepared sand/polymer composite sheets with water, benzene, water–benzene mixture, and printing ink. Contact angles for commercial stone paper and regular A4 paper are reported as well.

### 3.6. Printing Test

The printing tests showed the capability of the prepared sand/polymer composite sheets of absorbing printer ink. The sheets were printed on and left to dry for over 24 h. It was observed that some parts were still able to wipe off for all the prepared sand/polymer composite sheets, and some parts were able to stay on the surface. Pure HDPE showed the least absorption of the printing ink and was unable to retain the printing ink on the surface. The selected 35 wt% sheet prepared from 25 µm sand particles showed almost similar printability to its compatibilized version. However, for the case of the 35 wt% prepared from 5 µm sand particles, improved printability with the addition of the compatibilizer was noticed. Moreover, by observing the printing tests closely, ink had the ability to form drops yet not spread on the surface of the sand/polymer composite sheets prepared with either 25 µm sand particles or 5 µm sand particles, resulting in a blurred, non-graphic quality, which could be easily smudged. Similar results have been reported in the literature [13]. Figure 14 shows all the sheets after the printing tests (enclosed within the red highlights are the surfaces of the respective sheets).

On the stone paper, slight removal was observed after 24 h in contrast to the regular A4 paper, where the ink was completely dried and absorbed almost immediately. It can be concluded that the quality of the printing improved with the addition of the filler for the case of sand/polymer composite sheets prepared from 25 µm sand particles (compared to pure HDPE), and after compatibilization in the case of sand/polymer composite sheets prepared from 5 µm sand particles.

### 3.7. Adhesion Test

Selected sand/polymer composite sheet surfaces that were studied for the adherence test using a permanent marker pen are shown in Table 6. Amongst all the sand/polymer composite sheets, including the compatibilized sand/polymer composite sheets, the sand/polymer composite sheet prepared from 5 µm sand particles at 35 wt% showed relatively better adherence. Stone paper and regular A4 paper showed excellent adherence comparatively. 

Table 7 shows the sand/polymer composite sheet surfaces before and after the adherence test using removable marker pen. Only stone paper showed excellent adhesion compared to all the other tested sheets, which is reported in the literature as well [13]. For all the sand/polymer composite sheets prepared, the ink was easily removed with the tape irrespective of the particle size used or compatibilizer added. Moreover, the ink could be wiped out manually by swiping of the hand as well, which could be an advantage for applications that require such kind of erasable surfaces.

### 3.8. Comparative Analysis of Prepared Sand/Polymer Composite Sheets to Stone Paper

In this study, several properties varying from thermal to mechanical and wettability properties with several liquids were analyzed. Printing and adhesion tests were also tried on selected composite sheets as well as on the stone paper. To obtain a complete understanding of which sheets resulted in properties closest to the commercial stone paper, a comparative analysis was conducted.

In terms of percent crystallinity (%Xc), the compatibilized sand/polymer composite sheet prepared from 25 µm sand particles at 50 wt% yielded an almost similar value, of 41%, to stone paper, which resulted in 41.03%. With a peak degradation temperature (T_d_) of 490 °C, the sand/polymer composite sheet prepared from 25 µm sand particles at 50 wt% resulted in the closest value to stone paper, which had a peak degradation of 489.5 °C. In terms of mechanical properties, the compatibilized sand/polymer composite sheet prepared from 5 µm at 20 wt% resulted in an elastic modulus (E) of 603.54 MPa, whereas stone paper resulted in an elastic modulus of 596.32 MPa. For the tensile strength, with a value of 4.36 MPa, the compatibilized sand/polymer composite sheet prepared from 25 µm sand particles at 50 wt% gave the closest value to stone paper, which had a tensile strength (TS) of 6.17 MPa.

In terms of the wettability performance, the sand/polymer composite sheet prepared from 5 µm sand particles at 20 wt% gave almost the exact same contact angles (θ) with water to stone paper, each yielding value of 105.61° and 105.62°, respectively. The contact angle (θ) with printing ink that was closest to the stone paper (32.5°) was of the compatibilized sand/polymer composite sheet prepared from 5 µm sand particles at 35 wt% (37.55°).

The printing performance of the sand/polymer composite sheet prepared from 25 µm sand particles at 35 wt% showed the best results amongst all the tested composite sheets. In terms of the adhesion property using permanent marker, the sand/polymer composite sheets prepared from 5 µm sand particles at 35 wt% showed relatively better adherence compared to the other tested composite sheets. As for the adhesion property using a removable marker, all the tested sand/polymer composite sheets showed the same results. The sheets that resulted in values closest to the stone paper with respect to the analyzed properties are tabulated in Table 8.

## 4. Conclusions

In this work, sand/polymer composite sheets prepared from sand and HDPE were successfully manufactured via melt blending and compression molding at varying filler compositions. Thermal characterization revealed that the crystallization temperatures remained almost constant at ~113–115 °C, and the melting temperatures remained steady at ~132–135 °C for all the prepared sand/polymer composite sheets, promoting their good thermal stability. Moreover, no significant degradation (visual or on the bases of weight loss) at the optimized processing conditions were observed. The maximum degradation temperature was almost constant, ranging from ~489–493 °C for all the prepared sand polymer composite sheets, and similar crystallization, melting, and degradation temperatures were also obtained for the commercial stone paper, giving the prepared sand/polymer composite sheets decent thermal results. The mechanical characterization of all the sand/polymer composite sheets showed a decrease in their strength, as the elastic modulus values decreased significantly with the addition of the compatibilizer. The wettability analysis suggested that increasing the filler composition, as well as addition of the compatibilizer, led to an increase in the contact angles corresponding to the improved anti-wetting performance. Additionally, in the case of contact angles with printing ink, the observed angles were almost half the value obtained for the pure HDPE, varying in the ranges of 30–45°, which was also comparable to the stone paper, suggesting good potential for their ink wettability for printing and paper-based applications. The printing test showed that the prepared sand/polymer composite sheets gave comparatively better results than the pure HDPE sheets, implying some potential for printing with filler and compatibilizer addition. The prepared sheets showed good adherence with the use of permanent marker pen, suggesting potential for possible ink-based applications with such kind of ink. Furthermore, the sheets showed weaker adherence with a removable marker pen, but could possibly be used for any application that requires erasable surfaces (such as erasable sheets/boards for student learning, educational toys, etc.). The results obtained in this research provide information about the potential of the production of local sand/polymer composite sheets and their use in paper-based applications.

Key achievements:Local sand was used as a filler to develop composite sheets, which were investigated.Addition of a compatibilizer to the composite sheets was also investigated.Melt extrusion and compressing molding techniques were used.A set of two particle sizes of silica were used for this study.Data were compared to regular A4 paper, and stone paper was also analyzed.

## Figures and Tables

**Figure 1 polymers-14-03351-f001:**
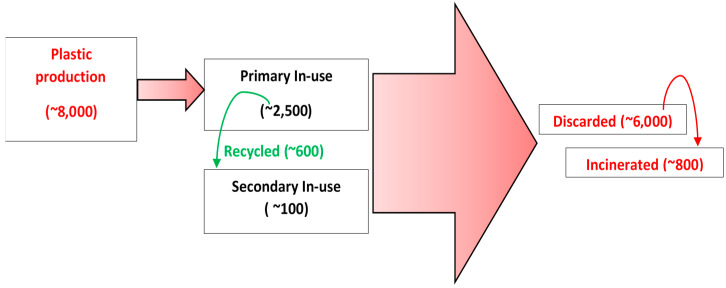
Simplified estimation of worldwide production and disposal of plastics in MMT for the last 65 years [9].

**Figure 2 polymers-14-03351-f002:**
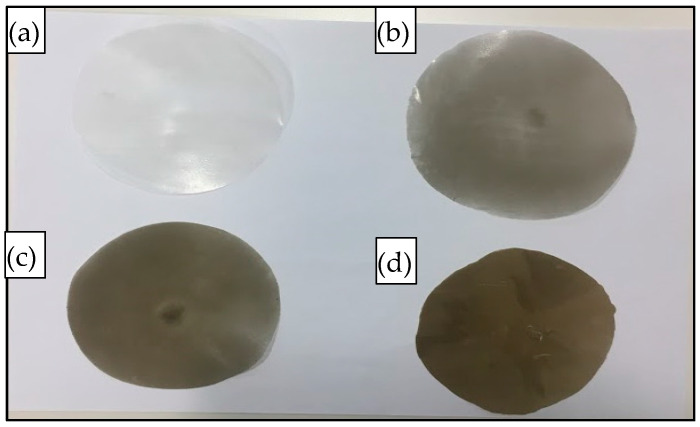
Images of prepared composite sheets for (**a**) pure HDPE, (**b**) 20 wt%, (**c**) 35 wt%, and (**d**) 50 wt% prepared from 25 µm sand particles.

**Figure 3 polymers-14-03351-f003:**
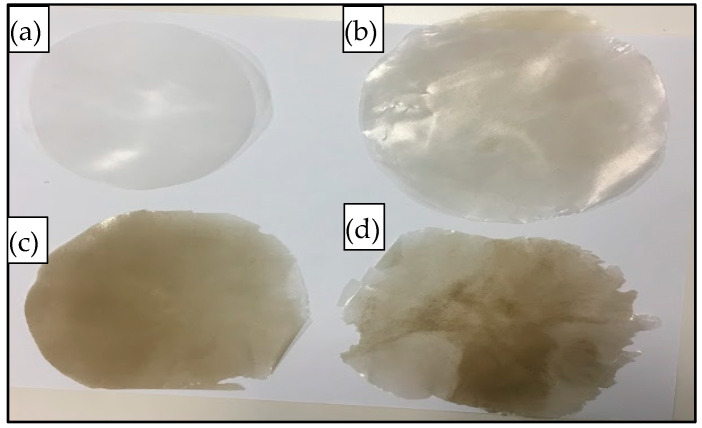
Images of prepared compatibilized composite sheets for (**a**) pure HDPE, (**b**) 20 wt%, (**c**) 35 wt%, and (**d**) 50 wt% prepared from 25 µm sand particles.

**Figure 4 polymers-14-03351-f004:**
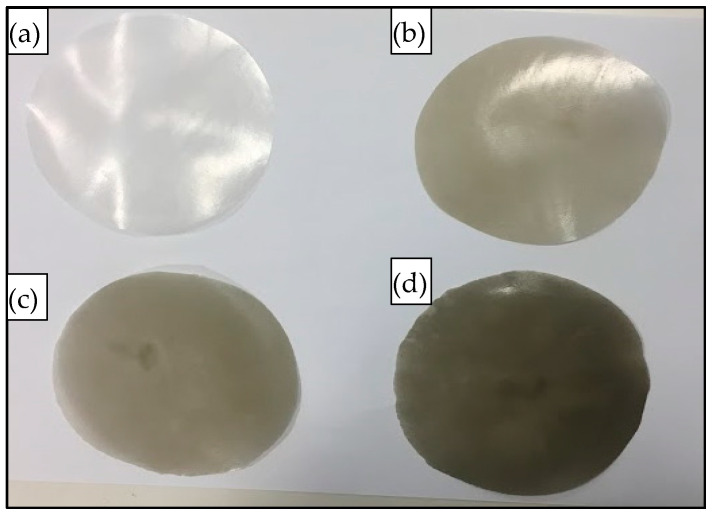
Images of prepared composite sheets: (**a**) pure HDPE, (**b**) 20 wt%, (**c**) 35 wt%, and (**d**) 50 wt% with 5 µm sand particles.

**Figure 5 polymers-14-03351-f005:**
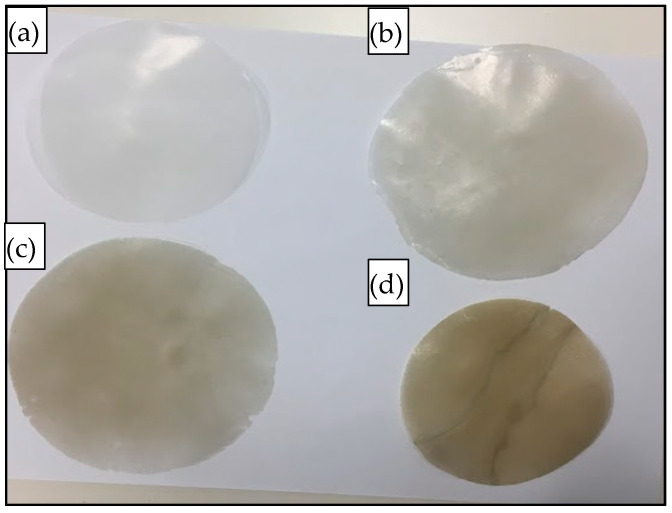
Images of prepared compatibilized composite sheets: (**a**) pure HDPE, (**b**) 20 wt% + C, (**c**) 35 wt% + C, and (**d**) 50 wt% + C, with 5 µm sand particles.

**Figure 6 polymers-14-03351-f006:**
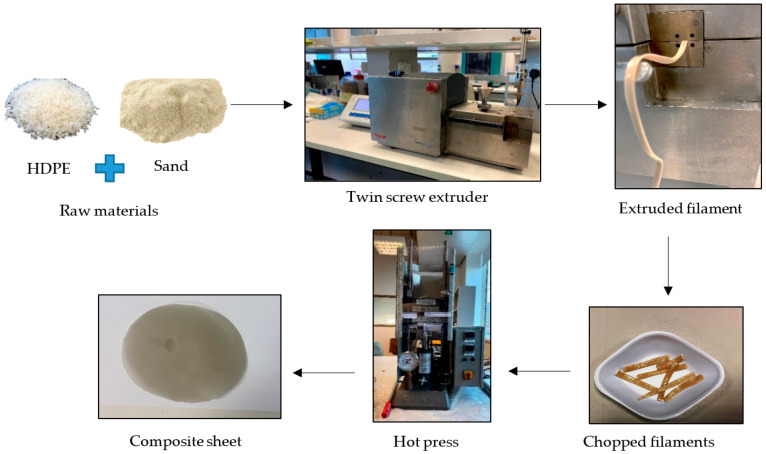
Process of preparing the composite sheets.

**Figure 7 polymers-14-03351-f007:**
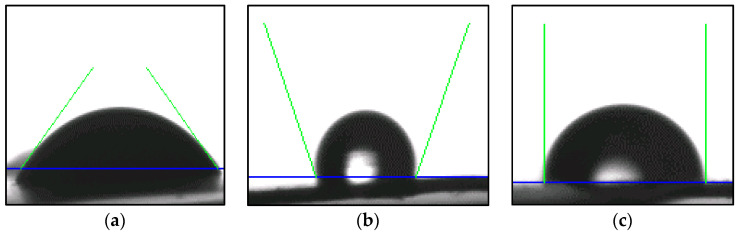
Contact angle measurements. (**a**) Extremely polar surface: (θ) = 90°, (**b**) Extremely non-polar surface (θ) => 90°, (**c**) Non-polar surface (θ) =< 90° taken via drawing tangent to the liquid droplet and the surface.

**Figure 8 polymers-14-03351-f008:**
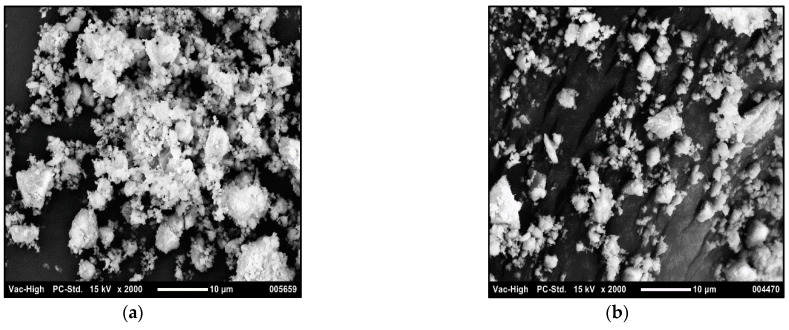
SEM images of (**a**) 25 µm sand particles and (**b**) 5 µm sand particles; (**c**) neat HDPE; (**d**) 50 wt%, 5 µm sand composite sheet; (**e**) 50 wt%, 5 µm sand compatibilized composite sheet.

**Figure 9 polymers-14-03351-f009:**
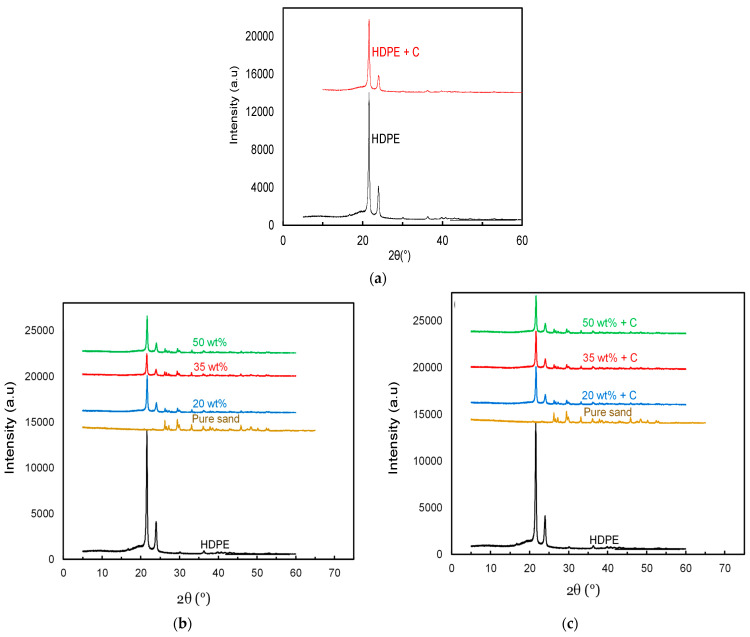
XRD characteristic peak patterns: (**a**) neat HDPE before and after compatibilization; (**b**) sand/polymer composite sheets prepared from 5 µm sand particles; (**c**) sand/polymer compatibilized composite sheets prepared from 5 µm sand particles.

**Figure 10 polymers-14-03351-f010:**
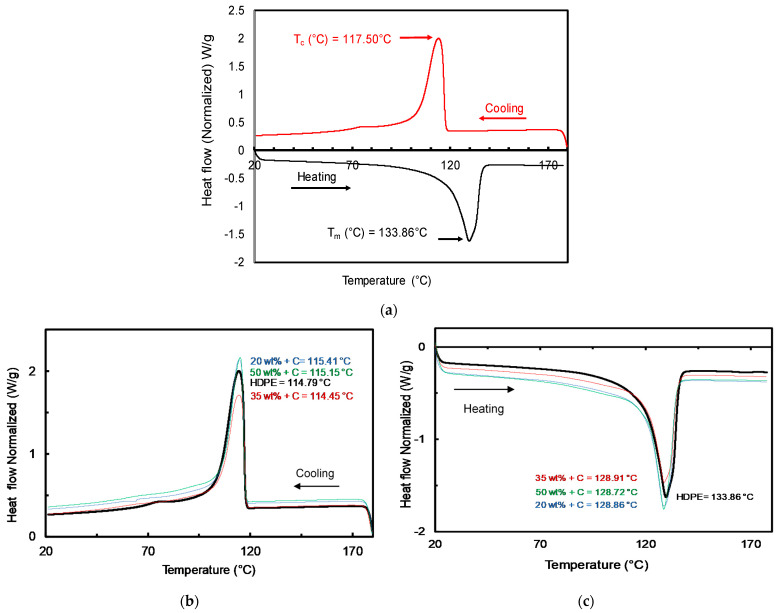
(**a**) DSC thermogram for neat HDPE. (**b**) Cooling profiles for compatibilized sand/polymer composite sheets prepared from 5 µm sand particles. (**c**) Heating profiles for compatibilized sand/polymer composite sheets prepared from 5 µm sand particles.

**Figure 11 polymers-14-03351-f011:**
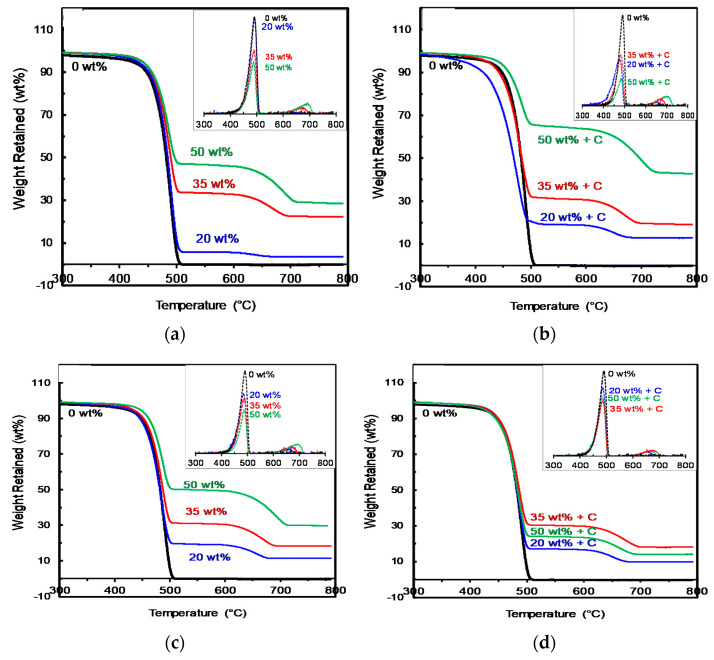
(**a**) TGA thermogram and DTG plot for sand/polymer non-compatibilized composite sheets prepared from 25 µm sand particles. (**b**) TGA thermogram and DTG plot for sand/polymer compatibilized composite sheets prepared from 25 µm sand particles. (**c**) TGA thermogram and DTG plot for sand/polymer non-compatibilized composite sheets prepared from 5 µm sand particles. (**d**) TGA thermogram and DTG plot for sand/polymer compatibilized composite sheets prepared from 5 µm sand particles.

**Figure 12 polymers-14-03351-f012:**
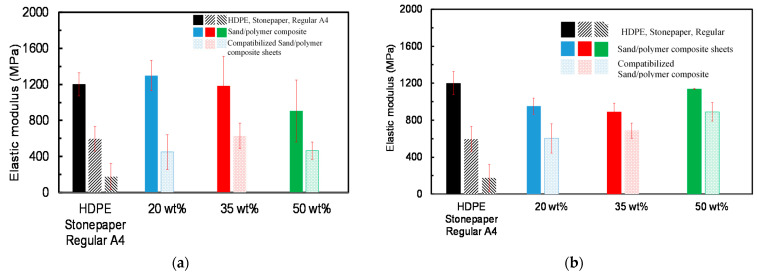
Elastic modulus for sand/polymer composite sheets and compatibilized sand/polymer composite sheets prepared from (**a**) 25 µm sand particles; (**b**) 5 µm sand particles.

**Figure 13 polymers-14-03351-f013:**
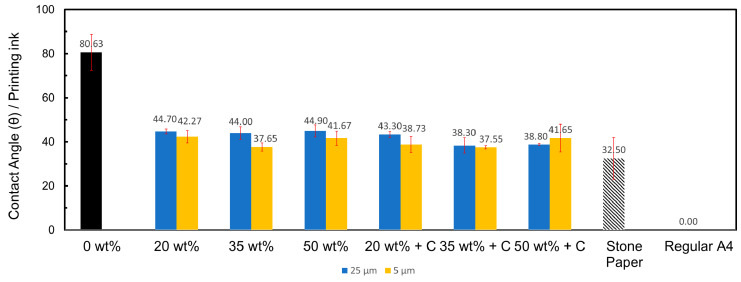
Contact angles of printing ink on sand/polymer composite sheets prepared from 25 µm sand particles and 5 µm sand particles.

**Figure 14 polymers-14-03351-f014:**
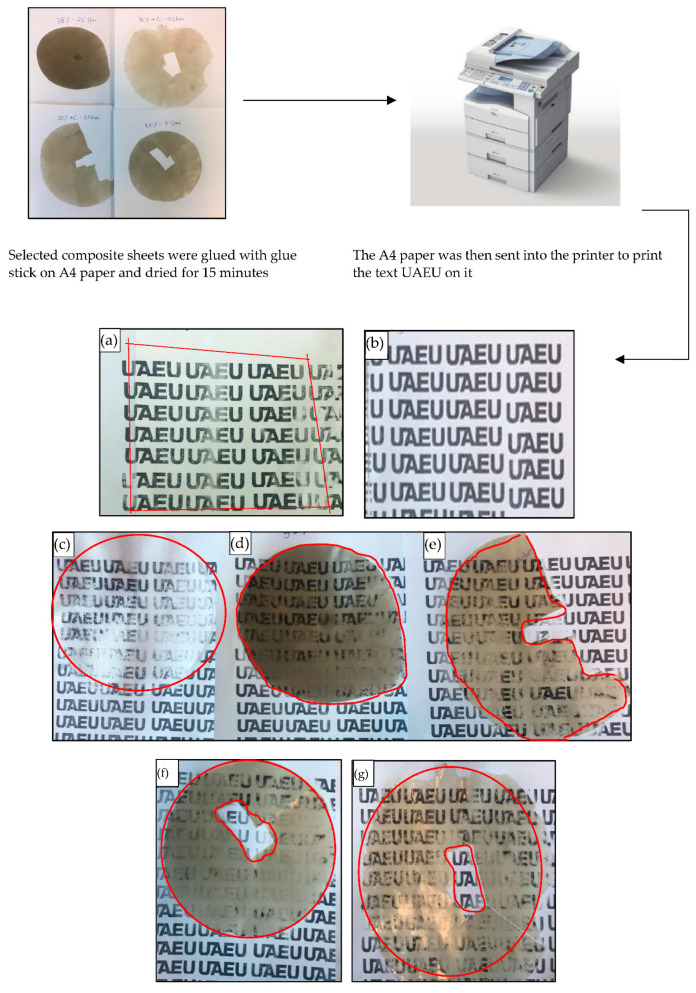
Printing tests after 24 h for (**a**) stone paper, (**b**) regular A4 paper, (**c**) HDPE, (**d**) 35 wt%, 25 µm (**e**) 35 wt% + C, 25 µm, (**f)** 35 wt%, 5 µm, and (**g**) 35 wt% + C, 5 µm.

**Table 1 polymers-14-03351-t001:** Chosen weight percentages for HDPE, sand, and compatibilizer.

	Sample	HDPE (wt%)	Sand (wt%)	Compatibilizer (wt%)
Sand/polymer composite sheets prepared from 25 µm and 5 µm sand particles	0 wt%	100	0	0
	20 wt%	80	20	0
	35 wt%	65	35	0
	50 wt%	50	50	0
	20 wt% + C	70	20	10
	35 wt% + C	47.5	35	17.5
	50 wt% + C	25	50	25

**Table 2 polymers-14-03351-t002:** DSC data for sand/polymer composite sheets prepared from 25 µm sand particles and 5 µm sand particles, stone paper, regular A4 paper.

	Sample	1st Cooling Scan			2nd Heating Scan		
		Tc,onset (°C)	Tc,peak (°C)	∆Hc (J/g)	Tm,peak (°C)	∆Hm (J/g)	%Xc
	0 wt%	117.54 ± 0.01	114.79 ± 0.61	120.62 ± 21	133.86 ± 0.55	148.77 ± 13.9	48.57 ± 8.5
Sand/polymer composite sheets prepared from 25 µm sand particles	20 wt%	117.46 ± 0.11	113.91 ± 0.83	121.51 ± 15	135.60 ± 3.5	131.27 ± 9.0	56.00 ± 3.8
	35 wt%	117.82 ± 0.06	114.61 ± 0.88	94.79 ± 14	135.423 ± 3.4	99.791 ± 9.0	52.40 ± 4.7
	50 wt%	118.71 ± 0.09	115.04 ± 0.33	74.38 ± 14	132.11 ± 0.07	86.034 ± 7.5	58.73 ± 5.2
	20 wt% + C	117.50 ± 0.07	115.40 ± 0.39	85.06 ± 9.1	133.58 ± 0.3	110.35 ± 2.5	47.08 ± 1.1
	35 wt% + C	117.82 ± 0.06	115.80 ± 0.40	87.03 ± 1.4	132.31 ± 1.9	96.60 ± 8.2	50.71 ± 4.3
	50 wt% + C	117.68 ± 0.49	114.95 ± 0.13	51.87 ± 16	134.01 ± 1.6	60.11 ± 5.4	41.0 ± 3.7
Sand/polymer composite sheets prepared from 5 µm sand particles	20 wt%	117.59 ± 0.08	115.04 ± 0.58	105.58 ± 10	133.35 ± 3.6	122.877 ± 11	52.42 ± 5.0
	35 wt%	118.23 ± 0.35	114.12 ± 1.32	98.32 ± 5	136.31 ± 4.2	109.07 ± 7.5	57.27 ± 3.4
	50 wt%	118.24 ± 0.05	116.21 ± 0.48	64.30 ± 6.2	133.72 ± 1.5	77.91 ± 5.7	53.18 ± 3.9
	20 wt% + C	117.66 ± 0.15	115.41 ± 0.64	112.23 ± 3.5	134.21 ± 5.1	118.77 ± 14	50.67 ± 6.3
	35 wt% + C	117.95 ± 0.21	115.48 ± 0.91	85.35 ± 3	134.56 ± 2.7	91.32 ± 2.5	47.95 ± 1.3
	50 wt% + C	117.78 ± 50	115.15 ± 0.16	42.49 ± 1.8	132.17 ± 2.7	54.51 ± 3.6	37.42 ± 3.4
	Stone Paper	117.72 ± 0.12	114.62 ± 0.1	27.08 ± 2.3	134.52 ± 0.8	33.26 ± 1.9	41.03 ± 3.7
	Regular A4	-	-	-	-	-	-

**Table 3 polymers-14-03351-t003:** TGA data for all the prepared sand/polymer composite sheets, stone paper, and regular A4 paper; 25 µm sand particles, 5 µm sand particles.

	Sample	T_d,onset_ (°C)	T_d,peak_ (°C)	T_onset, @ 10% weight loss_ (°C)
	0 wt%	466.69	489.0	444.86
Sand/polymer composite sheets prepared from 25 µm sand particles	20 wt%	458.03	491.0	448.85
	35 wt%	461.74	492.01	453.43
	50 wt%	464.6	490.0	454.66
	20 wt% + C	461.42	479.0	409.42
	35 wt% + C	460.11	481.3	439.66
	50 wt% + C	458.69	488.0	457.61
Sand/polymer composite sheets prepared from 5 µm sand particles	20 wt%	467.78	486.21	443.14
	35 wt%	464.31	490.53	449.48
	50 wt%	462.39	490.10	459.56
	20 wt% + C	437.68	499.01	446.97
	35 wt% + C	454.08	488.41	447.20
	50 wt% + C	456.76	488.06	442.17
	Stone Paper	457.05	489.50	469.72
	Regular A4	347.74	381.60	324.63

**Table 4 polymers-14-03351-t004:** Mechanical properties of all the prepared sheets, stone paper, and regular A4.

Sample		Elastic Modulus (MPa)	Yield Stress (MPa)	Yield Strain (MPa)	Tensile Strength (MPa)	Tensile Strain (MPa)
	0 wt%	1200.77 ± 127.3	35.15 ± 2.7	2.1 ± 3.5	33.76 ± 6.2	2.26 ± 0.42
Sand/polymer composite sheets prepared from 25 µm sand particles	20 wt%	1298.33 ± 169.8	27.84 ± 4.3	0.0424 ± 0.008	20.23 ± 3.5	0.17 ± 0.15
	35 wt%	1182.33 ± 328.4	23.11 ± 4.8	0.028 ± 0.009	18.96 ± 6.7	0.035 ± 0.01
	50 wt%	905.72 ± 343.1	12.74 ± 5.4	0.022 ± 0.009	9.93 ± 4.7	0.030 ± 0.02
	20 wt% + C	448.78 ± 194.1	11.98 ± 2.4	0.04 ± 0.01	8.01 ± 2.7	0.05 ± 0.01
	35 wt% + C	629.95 ± 138.9	9.56 ± 43.5	0.02 ± 0.004	8.68 ± 3.5	0.02 ± 0.005
	50 wt% + C	465.11 ± 94.9	6.71 ± 1.8	0.02 ± 0.01	4.36 ± 1.9	0.03 ± 0.01
Sand/polymer composite sheets prepared from 5 µm sand particles	20 wt%	950.59 ± 86.6	24.89 ± 1.9	0.04 ± 0.002	21.56 ± 0.6	0.06 ± 0.01
	35 wt%	887.47 ± 96.2	19.11 ± 0.9	0.03 ± 0.002	17.17 ± 2.8	0.05 ± 0.003
	50 wt%	1137.05 ± 8.2	18.64 ± 3.37	0.02 ± 0.002	17.22 ± 1.48	0.02 ± 0.001
	20 wt% + C	603.54 ± 157.9	17.64 ± 2.53	0.05 ± 0.02	15.94 ± 2.76	0.05 ± 0.02
	35 wt% + C	687 ± 80.87	212.48 ± 0.38	0.02 ± 0.00	10.98 ± 1.04	0.03 ± 0.01
	50 wt% + C	890.87 ± 101.6	16.78 ± 3.12	0.03 ± 0.005	15.09 ± 2.18	0.03 ± 0.02
	Stone Paper	596.32 ± 135.14	6.07 ± 0.53	0.11 ± 0.04	6.17 ± 0.73	0.54 ± 0.19
	Regular A4	175.18 ± 146.36	16.17 ± 5.08	0.06 ± 0.03	15.66 ± 5.14	0.06 ± 0.03

**Table 5 polymers-14-03351-t005:** Contact angle measurements.

Sample		Water	Benzene	Water-Benzene Mixture	Printing Ink
	0 wt%	97.96 ± 9.2	22.05 ± 1.8	100.93 ± 9.5	80.63 ± 8.2
Sand/polymer composite sheets prepared from 25 µm sand particles	20 wt%	86.62 ± 6.3	31.80 ± 2.6	90 ± 0	44.70 ± 1.1
	35 wt%	89.6 ± 0.9	22.83 ± 4.1	90 ± 0	44.0 ± 2.9
	50 wt%	94.72 ± 3.0	37.95 ± 6.9	90 ± 0	44.9 ± 2.8
	20 wt% + C	99.62 ± 6.2	26.03 ± 7.0	90 ± 0	43.40 ± 1.3
	35 wt% + C	92 ± 5.7	21.70 ± 6.9	90 ± 0	38.30 ± 3.4
	50 wt% + C	98.74 ± 8.3	20.17 ± 2.2	91.77 ± 3.1	38.80 ± 0.5
Sand/polymer composite sheets prepared from 5 µm sand particles	20 wt%	86.88 ± 5.5	30.95 ± 6.2	90 ± 0	42.27 ± 2.8
	35 wt%	92.08 ± 4.7	20.37 ± 2.1	90 ± 0	37.65 ± 1.9
	50 wt%	94.6 ± 8.6	27.03 ± 6.7	90 ± 0	41.67 ± 3.2
	20 wt% + C	105.6 ± 9.1	24.23 ± 0.8	90 ± 0	38.73 ± 3.7
	35 wt% + C	99.4 ± 3.9	21.13 ± 2.2	94.83 ± 8.3	37.55 ± 0.78
	50 wt% + C	99.12 ± 3.7	21.87 ± 1.6	102.33 ± 11.4	41.65 ± 6.3
	Stone Paper	105.62 ± 2.1	26.50 ± 4.4	113.30 ± 7.2	32.50 ± 9.5
	Regular A4	83.34 ± 7.8	0 ± 0	83.27 ± 21.8	0 ± 0

**Table 6 polymers-14-03351-t006:** Adherence test using permanent marker pen.

	Stone Paper	Regular A4 Paper	0 wt%	35 wt%, 25 µm	35 wt% + C, 25 µm	35 wt%, 5 µm	35 wt% + C, 5 µm
Before adherence test with permanent marker pen (sheet surface)	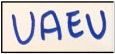	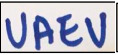	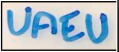	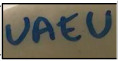	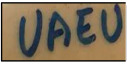	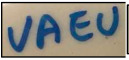	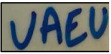
After adherence test with permanent marker pen (sheet surface)	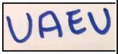	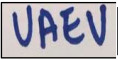	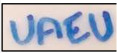	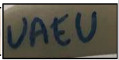	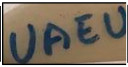	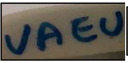	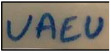
After adherence test with permanent marker pen (removed adhesive tape)	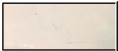	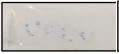	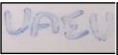	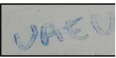	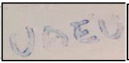	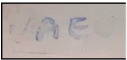	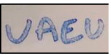

**Table 7 polymers-14-03351-t007:** Adherence test using removable marker pen.

	Stone Paper	Regular A4 Paper	0 wt%	35 wt%, 25 µm	35 wt% + C, 25 µm	35 wt%, 5 µm	35 wt% + C, 5 µm
Before adherence test with removable marker pen (sheet surface)	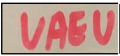	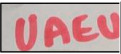	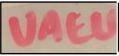	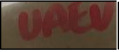	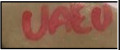	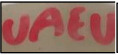	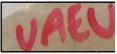
After adherence test with removable marker pen (sheet surface)	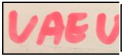	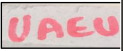	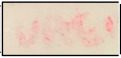	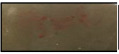	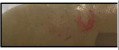	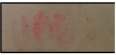	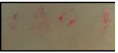
After adherence test with removable marker pen (removed adhesive tape)	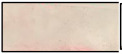	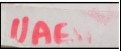	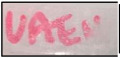	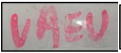	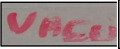	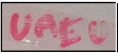	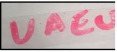

**Table 8 polymers-14-03351-t008:** Comparison of sheets resulting in values closest to stone paper.

Stone Paper		%Xc	T_d_	E	TS	θ/Water	θ/Printing Ink	Printing Test	Adhesion Using Permanent Marker
	Sheets
Sand/polymer composite sheets prepared from 25 µm sand particles	20 wt%								
35 wt%							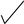	
50 wt%		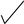						
20 wt% + C								
35 wt% + C								
50 wt% + C	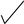			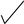				
Sand/polymer composite sheets prepared from 5 µm sand particles	20 wt%					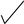			
35 wt%								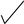
50 wt%								
20 wt% + C			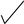					
35 wt% + C						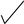		
50 wt% + C								

## Data Availability

Not required.

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
