# Peer review of "Development and Mechano-Chemical Characterization of Polymer Composite Sheets Filled with Silica Microparticles with Potential in Printing Industry"

_polymers, 2022, doi:10.3390/polym14163351_

Round 1

Reviewer 1 Report

Hello dear Authors,

Here are some comments to improve the quality of the paper.

·       The novelty of the work shall be explained in details.

·       More relevant works regarding using sand particles in polymers for different applications shall be added to the manuscript.

·       Some pictures of preparation process must be added.

·       Add the chemical properties of the used sand particles.

·       The maximum temperature during DSC thermal cycle was chosen as 180oC. Why did you decide to select this temperature.

·       What was the purpose of Thermogravimetric analysis (TGA) done in this research?

·       What tensile test crosshead speed was used?

·       It is written that “The rate of crosshead motion was set to be 100 mm/min, which was taken from the ASTM D 790 standard.” In one hand, this standard is used for flexural testing. On the other hand it seems like 100 mm/min test speed is too high for tensile testing.

·       The Printing test section is not clear. Add some images and explain the process in details.

·       The main achievement of the work must be shown in bullet in conclusion section.

Regards,

Reviewer 2 Report

The authors fabricated sand/polymer composite sheets and studied the mechano-chemical properties, and discussed the application of composite film in the field of printing. However, this manuscript needs a revision before it is considered for acceptance and publication. Here are the comments to the authors.

1.         Are the thicknesses of sand/polymer composite sheets measured by the authors?

2.         What is the dispersion of sand in the polymer matrix?

3.         In section “2.3.1. Scanning electron microscope (SEM)”, the author says “A JEOL/EO Scanning Electron Microscope (SEM) operated at 2 kV, spot size of 40 was used to image the sand of both 25 µm and 5 µm, neat HDPE, stonepaper and regular A4 paper. ” However, there are no SEM images of stonepaper and A4 paper in the manuscript.

4.         The authors need to confirm whether Figure 7c is an SEM image of HDPE or A4 paper. In addition, the authors need to specify whether the SEM image of the composite sheet is an image of its surface or a cross-section.

5.         Figures 6a and 6c are inconsistent with their captions.

6.         In the XRD patterns of Figure 8, the Y-axis was marked as arbitrary unit (a.u.), but there are some specific values in the Y-axis.

7.         To calculate the crystallinity of semi-crystalline polymer in a certain composite, it is necessary to know the mass ratio of the plastic to the composites. The authors need to explain how the crystallinity of the commercial "stonepaper" was obtained.

8.         The crystallinity of regular A4 was measured by DSC in the manuscript. The main components of A4 paper are cellulose, hemicellulose and lignin. Although these components are crystalline polymers, however, due to the specificity of the cellulose structure, their decomposition temperature is lower than the melting temperature, so it is almost impossible to measure the crystallinity of paper by DSC. The authors need to explain why regular A4 has a melting temperature and how the crystallinity could be calculated.
